# Safe Driving Distance and Speed for Collision Avoidance in Connected Vehicles

**DOI:** 10.3390/s22187051

**Published:** 2022-09-17

**Authors:** Samir A. Elsagheer Mohamed, Khaled A. Alshalfan, Mohammed A. Al-Hagery, Mohamed Tahar Ben Othman

**Affiliations:** 1Computer Engineering Department, College of Computer, Qassim University, Buraydah 52571, Saudi Arabia; 2Computer Science and Engineering Department, Egypt-Japan University of Science and Technology (E-JUST), New Borg-El-Arab City 21934, Egypt; 3Faculty of Engineering, Aswan University, Qism Aswan 81528, Egypt; 4College of Computer and Information Sciences, Imam Mohammad Ibn Saud Islamic University, Riyadh 11564, Saudi Arabia; 5BIND Research Group, College of Computer, Qassim University, Buraydah 52571, Saudi Arabia; 6Department of Computer Science, College of Computer, Qassim University, Buraidah 52571, Saudi Arabia

**Keywords:** Assured Clear Distance Ahead, Safe Driving Distance, Safe Driving Speeds, Internet of Vehicles, Intelligent Transportation Systems, smart cities, forward-collision avoidance

## Abstract

Vehicle tailgating or simply tailgating is a hazardous driving habit. Tailgating occurs when a vehicle moves very close behind another one while not leaving adequate separation distance in case the vehicle in front stops unexpectedly; this separation distance is technically called “Assured Clear Distance Ahead” (ACDA) or Safe Driving Distance. Advancements in Intelligent Transportation Systems (ITS) and the Internet of Vehicles (IoV) have made it of tremendous significance to have an intelligent approach for connected vehicles to avoid tailgating; this paper proposes a new Internet of Vehicles (IoV) based technique that enables connected vehicles to determine ACDA or Safe Driving Distance and Safe Driving Speed to avoid a forward collision. The technique assumes two cases: In the first case, the vehicle has Autonomous Emergency Braking (AEB) system, while in the second case, the vehicle has no AEB. Safe Driving Distance and Safe Driving Speed are calculated under several variables. Experimental results show that Safe Driving Distance and Safe Driving Speed depend on several parameters such as weight of the vehicle, tires status, length of the vehicle, speed of the vehicle, type of road (snowy asphalt, wet asphalt, or dry asphalt or icy road) and the weather condition (clear or foggy). The study found that the technique is effective in calculating Safe Driving Distance, thereby resulting in forward collision avoidance by connected vehicles and maximizing road utilization by dynamically enforcing the minimum required safe separating gap as a function of the current values of the affecting parameters, including the speed of the surrounding vehicles, the road condition, and the weather condition.

## 1. Introduction

Road traffic accidents kill approximately 1.3 million people each year, according to the World Health Organization (WHO), prompting the United Nations General Assembly (UNGA) to set an ambitious target of halving the casualties of road traffic accidents worldwide between now and 2030. One of the primary causes of traffic accidents is tailgating—tailgating consists in having a vehicle following another vehicle too close to brake safely if the followed vehicle stops unexpectedly, as shown in Figure 1).

To avoid tailgating, there is a need for a smart technique based on Intelligent Transportation System (ITS) and Internet of Vehicles (IoV) [1,2,3,4,5,6,7,8] to determine the ideal inter-vehicle space referred to as “Assured Clear Distance Ahead” (ACDA) [9,10] or Safe Driving Distance for connected vehicles on the smart roads and highways. Therefore, the existing ITS and Autonomous Emergency Braking (AEB) system [11,12,13] incorporated in smart vehicles require a good understanding of an ideal gap between connected vehicles [14] that will not result in a collision if the followed vehicle stops abruptly. Having a good knowledge of an ideal safe driving distance for connected vehicles will solve the following problems:(a)Forward collision due to tailgating.(b)Traffic optimization: leaving too much gap between two connected vehicles will result in poor utilization of the roads and highways and consequently create unnecessary traffic on crowded roads.(c)Over speeding (for the following vehicle) and under speeding (for the followed vehicle) on a busy highway.

The determination of ACDA by classical means is unsuitable for Intelligent Transportation Systems (ITS), smart cities, and smart roads [10]. ACDA depends on many variables, such as the vehicle’s speed, the length of the two vehicles, the condition of the tires, the type of road, and the weather condition; this makes the determination of the ACDA using traditional means very difficult.

IoV technology [1,2,3,4,5,6,7,8], as leading-edge technology, has been advancing for human’s need for a safe ITS, which provides a wealth of possibilities with most applications and services focusing on the safety of users and pedestrians [15,16,17]. IosV can be considered as the next generation ITS. Among the popular applications and services provided are automatic speed limit violation detection [18,19,20], traffic optimization and management [21,22], road lighting control and monitoring [4,23], transportation pollution monitoring [22,23], and infotainment [24].

In IoV, an intelligent vehicle (V) has an On-Board Unit (OBU) which extends the vehicle’s onboard computer’s capabilities. The simplicity of conversion of a non-IoV vehicle to an IoV vehicle can be done by incorporating the OBU into a non-IoV vehicle; this simplicity of converting a non-IoV vehicle into an IoV vehicle makes the deployment and the transition toward the next generation ITS easy.

IoV and Connected Vehicles (CV) consist of several entities [8,24,25]. The main entities are the intelligent vehicle (V), the road-side unit (RSU) (which forms the Infrastructure), the cloud servers (C), the smart grids (G) required to charge the Electric Vehicles (EV), the surrounding smart building (B), the home of the owner or the driver of the vehicle (H), the onboard sensors inside the vehicle (S), and the smart road traffic devices (R). The intelligent vehicle V may or may not be self-driving or driverless. There are several communications scenarios in IoV, such as V2V, V2I, V2S, V2P, V2C, V2H, V2B, V2G, and V2R; these scenarios govern the communication between the vehicle V and the Infrastructure (interconnected RSUs), the onboard Sensors S, the Pedestrians P using IoV-enabled wearables, the Cloud servers C, the Home H, the surrounding buildings B, the smart grids G, and the road devices R respectively [14,26]. For instance, Intelligent vehicles (V) communicate with one another using V2V (Intelligent Vehicle to Intelligent Vehicle) communications.

IoV is a promising technology enabling the deployment and development of endless practical applications in smart cities [4,27,28,29,30,31]; it is considered the next generation ITS, enabling safety, traffic optimization, and comfort applications.

In this paper, we are setting to find the Ideal Safe Driving Distance (and hence Optimal Safe Driving Speed) based on IoV for the avoidance of forward collision due to tailgating so that other researchers and automobile manufacturers can use this research as a background for developing applications that require the knowledge of the ideal gap between two vehicles. The knowledge can also be used to assess safe driving or detect safe driving violations.

We can summarize the contributions of this paper as follow:(1)We studied the different measurement techniques that can be used to measure in real time the Assured Clear Distance Ahead (ACDA) in the IoV environment and autonomous vehicles.(2)We have studied in detail all the factors affecting the Stopping Distance by studying the braking dynamics in the IoV and the autonomous vehicles.(3)We conducted a complete study considering all the parameters affecting the safe following distance and speed as the current speed of the followed vehicle, the speed of the following vehicle, the separation distance, the deceleration, the driver reaction time, road conditions (Asphalt, pavement, wet, dry, snow), the weather conditions (rainy, foggy, clear), the mass of the vehicles, the braking force, the tires state, etc.(4)Studying the effect of using the Autonomous Emergency Braking (AEB) system that exists in some vehicles and is a must in autonomous vehicles. In other words, we studied the case when the rear vehicle auto-brake in case of emergency, hence eliminating the driver’s reaction time, especially in foggy weather or bad visibility conditions, hence maximizing the road efficiency and decreasing the trip time.(5)Studying the case when the followed vehicle instantly stops and the effect of the safe driving distance and the safe following speed. In other words, we studied the effect of the sudden stop of the followed vehicle on the safe driving distance and speed for the different conditions.(6)We formulated how to use the IoV emergency safety message to exchange the related parameters between the followed and the following vehicles so that each vehicle on the road can calculate the ideal safe following distance and speed according to the current conditions (such as road conditions, weather conditions, car conditions, vehicle’s locations and speed, the length of the vehicles, etc.).

The rest of the paper is organized as follows. In Section 2, related works are presented. The various tools and techniques for measuring ACDA are covered in Section 3. In contrast, analysis of the technique we proposed for determining Ideal Safe Driving Distance under different variables is discussed and analyzed in Section 4. Finally, the conclusions, future work directions, and fund statements are covered in Section 5.

## 2. Related Works

Several previous works tackled safe driving distance determination. Table 1 summarizes the important related studies. In [32], the safe driving distances at the intersection and straight roads were considered a function of speed and deceleration. The study did not consider many important parameters such as the road stat, the current separation gap, the tires condition, the visibility, the weather conditions, the weight of the vehicles, and the braking force. The study did not consider the case when the front vehicle stops instantly (in zero time). In addition, the study did not consider the effect of different driver reaction times. Additionally, the study did not consider the cases when the vehicle is equipped with an Autonomous Emergency Braking (AEB) system or not. The study did not consider the different types of distance measurement techniques used in IoV and CV.

In [33], a distance estimation algorithm considering the road slopes is proposed. The study does not consider any parameter of those considered in our proposed work. The work proposes an algorithm that uses a single camera to estimate the distance; it does not consider the safe driving distance or the safe driving speed.

The study presented in [36] provides a safe following driving distance in different fog levels. The study does not consider any of the other important conditions that we are considering our study. A similar study considering only the fog condition is given in [37]. The study focuses only on the daytime foggy areas.

Simulation and reinforcement learning to determine the safe driving distance as a function of the speed of the following vehicle and the separation gap only was given in [38]. A similar study considering the safe-following speed using reinforcement learning is given in [39]. Another study considering the use of reinforcement learning is given in [41]. The study takes into consideration the used energy.

In [40], the authors present a stereovision-based approach for determining the safe driving distance. The proposed approach consists of having two cameras mounted on the security vehicle. The distance between the security vehicle and the ahead vehicle can be calculated using traditional camera calibration, matrix projection, and parameter distortion calculation. Although this approach is effective, it requires the presence of a security vehicle, which can be noticed by the driver. Furthermore, it is not suitable for next-generation ITS and connected vehicle technologies. In [42], the authors propose using the backward shockwave analysis technique for predicting active safe driving in ITS based on cloud computing analysis of big data in unstable driving conditions.

The authors of [34] proposed a system where vision sensors collect data by recording and analyzing images in a stereoscopic fashion by rear cameras in order to measure the distance between the leading and trailing vehicles. Visual data relevant to the safety distance is transmitted in real-time to the following vehicle using an asynchronous collaborative procedure. A complete error analysis of the distance computation is presented based on the measuring process and highway geometry; this innovative and cost-effective technology can recognize vehicles and deliver a rear-end distance warning system and also alert other vehicles of the danger if the minimum safe distance is violated

Unlike the time-headway and traditional braking models, a new safety indicator called time gap interval for safe following distance (TGFD) is proposed in [26], which incorporates vehicle dynamics and driver behavior factors, including the time component, to broadcast and propagate appropriate safety messages in a vehicular ad hoc network (VANET) environment.

Based on reinforcement learning, a model of velocity control during automobile following is developed [38]. A reward function is created by referencing human driving data and integrating safety, efficiency, and comfort characteristics. Compared to the MPC-based ACC algorithm, the suggested model outperforms it in terms of safety, comfort, and, most importantly, running speed during testing (more than 200 times faster). The findings suggest that the suggested method might aid in developing improved autonomous driving systems. In [43], Video data are processed with photogrammetry to evaluate if a vehicle is keeping a minimum safe distance or ACDA from the car in front of it, based on the travel speed. The device automatically calculates the speed and distance between cars and detects any relevant breaches. The system may be configured to cover up to four traffic lanes, with a single camera used to identify infringement.

To explain the microscopic car-following behavior of RVs (Regular Vehicles) and AVs (Autonomous Vehicles) [44], offers a mixed-vehicle car-following model based on the FVAD (Full Velocity and Acceleration Difference) model. The velocity of several front vehicles and a rear vehicle, as well as the velocity difference, acceleration difference, and headway between each front vehicle and the host vehicle, are all included in the model. A stereo-vision-based pedestrian identification and collision-avoidance system for AVs is proposed in [45]. In this system, the minimum safe distance is 3.3 m. If the anticipated distance is less than 3.3 m, the Autonomous Emergency Braking (AEB) system controller algorithm will activate the AEB. The technology estimates the distance once a pedestrian is identified. The implementation is done in MATLAB, and the experimental results show that the suggested strategy is promising in prediction accuracy and decreasing fatalities. However, since the minimum safe driving distance or ACDA dependents on several parameters, 3.3 m cannot be reliable.

The maximum distance of the highway ahead visible to the motorist is referred to as the available sight distance (ASD) is examined in [46]. Road geometry principles are a fundamental aspect of road planners employing them to provide safe driving conditions. In another similar work [47], a driving simulator study was carried out to determine a reasonable speed limit and ensure traffic safety in a dynamic low-visibility environment with fog. The combined effect of visibility and driving speed on drivers’ recognition time was investigated. Based on the stopping sight distance model, a method for determining and recommending a reasonable driving speed limit was proposed.

Lastly, in [48], the study first establishes the car-following and overtaking model based on traffic laws of driving in the right lane unless overtaking occurs. In the car-following model, the traffic features are analyzed to obtain the function relationship between the safety distance and speed change; in the overtaking model, the TWOPAS simulation model is used to analyze the highway traffic capacity and obtain the speed-flow relation graph, and the overtaking ratio-flow relation scatters graph.

## 3. Assured Clear Distance Ahead (ACDA) Measurements

Assured Clear Distance Ahead (ACDA) is the separation between the followed vehicle and the following vehicle that will not cause a crash if the followed vehicle stops unexpectedly; this section provides state-of-the-art techniques and technologies used to measure or estimate the separation distance between the followed and the following vehicles.

Intelligent vehicles are typically equipped with different types of distance measurement sensors, as shown in Figure 2. Table 2 shows the distance measuring technologies used in vehicular systems; it can be observed that Standard GPS, GNSS, Differential GPS (DGPS), and Real-Time Kinematic (RTK) positioning have the maximum range (unlimited), and the Infrared Proximity sensor has the least (1.5 m). In terms of accuracy, RTK and Infrared Proximity Sensor have the best rating (±1 cm), and RTK also has the minimum range (0.01 m). However, the Update Rate of Micro/Short LiDAR and Long-Distance LiDAR (1–1000 Hz) makes the two technologies popular.

Several approaches for measuring the distance between the current vehicle and the front vehicle using stereo vision cameras are proposed in [34,40,45,52], The stereo-vision system is a computer vision system that calculates distance using stereoscopic ranging algorithms; this technology uses two cameras as one, attempting to create the illusion of depth and calculating the distance with high precision based on the disparity of the objects between the cameras. As shown in Figure 3, the distance to the front vehicle can be estimated using two cameras mounted on the vehicle separated by a known distance. The reported accuracy is around 3 cm. Stereo vision can be used for distances of up to 200 m [53], depending on the type of camera used.

Figure 4 shows the Radio Detection and Ranging (RADAR) distance measurement. The distance to an object is calculated by monitoring the reflection of a high-frequency signal from that object. RADAR uses radio waves to calculate an object’s distance (ranging), angle, or velocity, which makes it useful for things like fluid level reading, traffic distance, and object detection [54].

Figure 5 shows Light Detection and Ranging, or LiDAR, which has recently been used in intelligent vehicles. The OBU of an intelligent connected vehicle is equipped with several real-time location systems making IoV vehicles determine their current location [55]. The Global Positioning System (GPS) is one of the most widely used systems [51]. Many issues, including accuracy, limit the use of GPS in various IoV applications [56]. If a device can receive location information from multiple satellite systems, it can significantly improve average accuracy. The accuracy, the speed, and the extended range of LiDAR make it popular in IoV-enabled vehicles.

Given that each vehicle in IoV knows its absolute real-time location, the distance between any two vehicles could be calculated as follows. First, each vehicle periodically broadcasts its current location while it is moving. Each vehicle receives the broadcast locations of all surrounding vehicles. The vehicle can use GNSS or RTK to calculate its moving direction (heading angle) and broadcast this information. Thus, any vehicle can know which vehicles are moving in the same lane and direction. The vehicle calculates the distance to all vehicles that are in the same lane and then selects the shortest distance of the vehicles that are moving ahead. Thus, the vehicle can know the distance to the followed vehicle. In tunnels and city centers where the GNSS signals are not received, several other localization techniques are used, such as the received signal strength from different radio sources (RSU, for example) [57,58,59].

## 4. Stopping Distance and Braking Dynamics

It is essential to understand the braking dynamics to be able to estimate the safe driving speed and the safe driving separation distance or ACDA. In this section, we present the Stopping Distance (SD) affecting parameters and the braking dynamics. As shown in Figure 6, when the driver perceives a danger, he/she takes some time before making the decision to brake; this is known as the thinking time, $t_{1}$. Once the driver decides to brake, it takes a period to move his/her leg and press on the brake pedal; this period, $t_{2}$, is called the reaction time. Usually, the braking system does not engage instantaneously, but it takes a few milliseconds to fill up the braking system to create the necessary brake compression; this period is called the brake effectiveness time, $t_{3}$. During all these periods, the vehicle moves with a constant speed. Once the brake becomes effective, the vehicle decelerates at a constant deceleration, j,  until it stops completely; this period is called the braking period, $t_{4}$. The stopping time $T_{s}=t_{1}+t_{2}+t_{3}+t_{4}$. According to [26], if the driver is alert, then $t_{1}=0.5$ s, $t_{2}=0.2$ s, and $t_{3}=0.3$ s. Nowadays, some drivers are not alert (e.g., using his/her smartphone while driving), in such a case, the perception time may be more than 1.0 s.

The braking time $t_{4}$ is affected by different factors such as the braking force, tire state, tire type, and friction force.
(1)$$SD=V.\left(t_{1}+t_{2}+t_{3}\right)+\frac{V^{2}}{2.g.\;\left(f\pm s\right)}$$
where *V* is the vehicle speed, $t_{1}$ is the thinking time, g=9.81 is the gravity constant, f is the adhesion coefficient that varies with the road type, as shown in Figure 7, and s is the slope of the road. Figure 7 shows the adhesive coefficient (*f*) for different road types/conditions; it can be seen that Asphalt (Dry) has the highest adhesive coefficient (*f* = 0.9) whereas Icy road has the lowest adhesive coefficient (*f* = 0.1). There is also a 10% decrease in the adhesive coefficient of Asphalt (Dry), Pavement (Dry), Asphalt (wet), and Pavement (wet), respectively. An increase in adhesive coefficient results in a sharp decrease in braking distance, as shown in Equation (1).

The tire state significantly affects the braking distance. Figure 8 shows the effect of tire state on the braking distance of a moving connected vehicle, provided the speed and nature of the road are constant. As it can be seen, with new tires (8 mm tread), the braking distance is at a minimum (60 m), and worn-out tires (3 mm tread) result in 90 m braking distance (50% increase in braking distance) whereas another surge in braking distance (115 m) is observed when using bald tires with 1.5 mm tread. Deductively, the braking distance increases as the tires wear out over time.

Table 3 presents the total stopping distance (in meters) for different road types and for different speeds. As can be seen, Asphalt (Dry) has the least total stopping distance at the lowest speed (3.2 m at 10 km/h) and the lowest stopping distance at the highest speed (138 m at 150 km/h) compared to the other road types. Deductively, the icy road has the longest total stopping distance at the slowest speed (6.7 m at 10 km/h) and the highest stopping distance at the fastest speed (927 m at 150 km/h) compared to Snowy roads and others.

Figure 9 presents the extra stopping distance of vehicles traveling on dry and wet asphalt at different speeds. The extra stopping distance for a vehicle traveling on wet asphalt compared to dry asphalt is 1.2 m at 30 km/h. However, the extra stopping distance becomes longer at higher speeds. For example, at 100 km/h, the total stopping distance on dry and wet asphalt is 71 m and 84 m, respectively, yielding an extra stopping distance of 13 m. Moreover, the extra stopping distance at 150 km/h is 30 m.

The extra stopping distance in meters on snowy roads compared to dry asphalt roads for different speeds is presented in Figure 10. For example, at 30 km/h, the stopping distance of a vehicle traveling on dry asphalt is 12.2 m compared to 26.0 m on a snowy road at the same speed by the same vehicle. That is 13.8 m extra stopping distance (more than the stopping distance in dry asphalt). Moreover, the extra stopping distance is even more at 150 km/h, where the stopping distance on dry asphalt is 138 m, compared to 484 m on a snowy road (a considerable difference of 346 m, an increase in extra stopping distance of more than 250%).

In Figure 11, the stopping distance as the sum of reaction distance and braking distance for different speeds on different road types is presented, assuming the reaction time is 1 s assuming the vehicle has new tires. Three cases are shown as:
CASE A: The stopping distance of a vehicle traveling on dry asphalt road at different speeds as reaction distance plus braking distance. At a low speed (for example, 30 km/h), the reaction distance is doubled when the braking distance 8 m (independent of the type of road), and the braking distance is 4 m. The braking distance becomes higher at high speed (braking distance = 84 m compared to a reaction distance of 39 m at 140 km/h).CASE B: On a snowy road, the stopping distance is longer mainly due to the longer braking distance. At 20 km/h, for example, the braking distance is 58.5% of the stopping distance (7.9 m), and at higher speed, for example, at 140 km/h, the braking distance is 90.8% of the stopping distance (385 m).CASE C: The same phenomenon is observed on icy roads. The braking distance at 20 km/h is 73.7% (15.7 m) of the stopping distance and at a higher speed of 140 km/h, the braking distance is 95.2% of the stopping distance.

Table 4 presents the stopping distance (in meters) as the sum of the reaction distance (in meters) and braking distance (in meters) for different road types. As we showed from the previous data obtained, the reaction distance is independent of the road type since the reaction time is the same regardless of the road type the driver is traveling on. Dry asphalt has the least braking distance at the same speed as the other road types, while the icy road has the highest braking distance and hence the highest stopping distance at the same speed. At 20 km/h, the braking distance on dry asphalt is just 1.7 m compared to dry pavement (2.0 m), wet asphalt (2.2 m), wet pavement (2.6 m), snowy road (7.9 m), and Icy road (15.7 m). There is a similar trend even at higher speeds. For example, at 140 km/h, dry asphalt has a braking distance of 84 m compared to dry pavement, wet asphalt, wet pavement, snowy road, and icy road, which have 96 m, 110 m, 128 m, 385 m, and 771 m respectively.

### 4.1. Sufficient Safe Gap between Two Vehicles

In the previous section, we presented the stopping dynamics when the driver of one vehicle perceives danger and decides to stop. We assumed that the vehicle was moving with a constant speed and then constant deceleration until it stops completely. However, vehicles are moving normally at approximately the same speed. If two vehicles realize the same danger at the same time and start barking at the same time, they will not collide as they will stop at the same time, maintaining the same gap while they brake. If we consider the stopping distance presented in the previous section as the minimum safe driving distance or Assured Clear Distance Ahead (ACDA), this will reduce road utilization efficiency and hence create many traffic management and optimization problems. Without IoV technology, it is not possible to objectively obtain ACDA for each vehicle as a function of the speed of the two vehicles, the tire states, the road type and condition, the weather condition, etc. In this section, we aim to calculate the sufficient ACDA dynamically, taking into account all affecting parameters when the two vehicles are moving at different speeds and decelerate with different decelerations.

We show in Figure 12 two vehicles A and B are moving, keeping a gap distance, *d,* between them at a specific time t. Let’s assume that the rear vehicle (A) is moving at a speed$\;V_{A}$, located at $L_{A}$, having a tire state $T_{A}$, and the length of the vehicle is $l_{A}$. The front vehicle (B) has its own parameters too. The two vehicles exchange these parameters in real-time periodically using V2V communication. If the front vehicle perceives danger and decides to brake, this decision is instantly sent to all nearby vehicles, including the rear vehicle A. Vehicle B decelerates with a deceleration $j_{B}.$ Thus, vehicle A starts braking using a deceleration $j_{A}$. If we assume that the stopping distance of both vehicles are $S_{A}\;\mathrm{and}\;S_{B}$, then the distance between them when they stop will be
$$\delta =S_{B}-S_{A}+d$$

When the front vehicle B starts braking, the rear vehicle A will not brake instantly, it will continue at its current speed during the thinking and the reaction times (as shown in Figure 6). During that time, the vehicle traveled a distance $V_{A}.\;T_{rA}$, then it will decelerate.

The work presented in [48] aims to estimate the safety gap to avoid rear-end collision safe running on highways. The authors propose three formulas for calculating the safety gap. The proposed formulas consider the speed and acceleration/deceleration of the two vehicles to estimate the sufficient safety gap to avoid a rear collision if the front vehicle brakes suddenly. We will use the proposed formulas in the next section as the basis of developing an enhanced one taking into consideration different realistic parameters such as the weather condition.

We must calculate the braking distance of the followed vehicle *B*: $\frac{V_{B}^{2}}{2gf}$.

Stopping distance of *A* (following vehicle) = *D* + Braking Distance of *A* (followed vehicle), neglecting the reaction time of *A*.
(2)$$V_{A}.\;T_{rA}+\frac{V_{A}^{2}}{2gf}=\frac{V_{B}^{2}}{2gf}+D$$

For any given initial distance between the two vehicles, *f* and $V_{B}$ we can find the maximum speed that can provide the minimum safe distance between the two vehicles by solving the previous quadratic equation, yielding
(3)$$V_{Amax}=\frac{-2fgT_{rA}+\sqrt{{\left(2fgT_{rA}\right)}^{2}+4\left(V_{B}^{2}+2gfD\right)}}{2}=\sqrt{{\left(fgT_{rA}\right)}^{2}+V_{B}^{2}+2gfD}-fgT_{rA}$$
(4)$$D_{min}=V_{A}.\;T_{rA}+\frac{V_{A}^{2}}{2gf}-\frac{V_{B}^{2}}{2gf}$$

### 4.2. Weather Effects on the Stopping Distance

Figure 13 shows the effects of weather conditions and tire type on the stopping distance of a vehicle. A vehicle with a winter tire traveling at a speed of 100 km/h on wet roads needs a 66 m intervehicle gap and 71 m with summer tires. Meanwhile, on icy roads, traveling at a speed of just 30 km/h needs a stopping distance of 57 m (with winter tires) and 68 m with summer tires.

We have calculated the maximum speed of the following vehicle for different speeds of the followed vehicle given an initial gap. In the case of dry asphalt, which is the ideal condition, the maximum speed of vehicle A (*V_Amax_*) and the initial gap between the vehicles assuming the reaction time of the driver of 1 s are shown in Figure 14. At around 30 km/h, the initial gap between the vehicles is 10 m, and the maximum speed of almost 180 km/h with a 100 m gap between the vehicles. However, in the case of snow, the friction coefficient becomes very low, and thus the stopping distance increases, which affects the initial ideal gap between the two vehicles. The visibility significantly affects the stopping distance [36,37]. Thus, in foggy weather, the driver’s reaction time is usually increased from 1 s to 8 s [36,37,47,60].

We calculated the same values in the case of foggy weather, as shown in Figure 15. As it can be seen, the increase in the driver’s reaction time significantly increases the stopping distance; hence, the maximum allowed speed of the following vehicle must be decreased. For example, if the followed vehicle is traveling at a speed of 100 km/h and the initial separating distance with the following vehicle is 100 m, then the maximum allowed speed of the following vehicle is 150 km/h in the case of dry asphalt in clear weather while it is only 58 km/h in the case of dry asphalt in foggy weather. When the road condition is bad, such as in the case of snowy roads, the maximum speed in the case of clear weather is 118 km/h, while it is dropped to 76 km/h in the case of foggy weather.

Similarly, we calculated the minimum safe driving distance for different speeds of both the following and the followed vehicle when the weather is clear, assuming the driver reaction time is 1 s, and when the weather is foggy, assuming that the driver reaction time is 8 s. Figure 16 and Figure 17 show the obtained results in the two cases, considering the dry asphalt and snowy roads. As shown, the minimum distance between the vehicles is proportional to the speed of vehicle B (the followed vehicle) on both road types, but the minimum distance is longer on snowy roads. For example, the minimum safe driving distance in the case of dry asphalt when the followed vehicle travels at a speed of 120 km/h and the following vehicle travels at a speed of 100 km/h is 52 m and 280 m in the cases of clear and foggy weathers respectively. In the case of snowy roads, for the same conditions, the minimum allowed safe driving distance is 130 m and 370 m, respectively.

### 4.3. Case When the Rear Vehicle Auto-Brake in Case of Emergency

We have two cases: the case if the vehicle’s brake is controlled by the driver only and the case if the vehicle OBU triggers the braking system automatically in the case of any detected emergency. In the latter case, the followed vehicle detects an emergency situation, and when its brake is pressed, it will immediately send an emergency brake warning message to the surrounding vehicles using V2V. Thus, the following vehicle OBU receives this message and automatically triggers the braking system. The time for sending and receiving this message is in the order of a few milliseconds, which can be neglected. Recently, several vehicles have been equipped with an Autonomous Emergency Braking (AEB) system that brakes the vehicle autonomously in the case of perceived danger without any intervention from the driver. Connected vehicles may have this system, while autonomous vehicles must have this system.

In the case when the vehicle can brake instantly when it receives an Imminent-Danger message from any of the vehicles in front of it using V2V, then the driver reaction time of the following vehicle is zero or $T_{rA}=0$. In such a case, we have the following equations:(5)$$\frac{V_{A}^{2}}{2gf}=\;\frac{V_{B}^{2}}{2gf}+D$$
(6)$$V_{Amax}=\sqrt{V_{B}^{2}+2gfD}$$
(7)$$D_{min}=\frac{V_{A}^{2}}{2gf}-\frac{V_{B}^{2}}{2gf}$$

In such a case, the maximum driving speed of the following vehicle will be increased, and the minimum safe driving distance given the speed of the two vehicles will decrease. Figure 18 presents the maximum speed of the following vehicle for different speeds of the followed vehicle for different initial distances between the two vehicles when the following vehicle is dotted with an Autonomous Emergency Braking (AEB) system ($T_{rA}=0\;s)$ for both dry asphalt and snowy roads. For example, when the followed vehicle is traveling at a speed of 100 km/h and the current separation distance between the two vehicles is 100 m, the maximum safe driving speed of the following vehicle is 185 km/h and 125 km/h in the case of dry asphalt and snowy roads respectively. If we compare these values with those obtained for the same conditions when there is a driver reaction (clear weather), the maximum allowed speed is 150 km/h and 110 km/h (as shown in Figure 14). As it can be seen, this contributes to maximizing road utilization, reducing traffic congestion, and decreasing pollutant emissions.

Similarly, we calculated the minimum safe driving distance for different speeds of both the following and the followed vehicle in the case when the following vehicle is dotted with an Autonomous Emergency Braking (AEB) system ($T_{rA}=0\;s)$ for dry asphalt and snowy roads. The results are presented in Figure 19. As can be seen, when comparing the results depicted in this figure with those depicted in Figure 15 using the AEB significantly decreases the minimum safe driving distance for the same values of the driving speeds of both vehicles.

### 4.4. Sudden Stop of Vehicle B

In some cases, the followed vehicle may stop suddenly (its speed drops to zero in zero seconds), in the case of a sudden crash with a stopped vehicle, for example, or in any other critical situation. If ACDA is calculated according to what has been described in the previous subsections, the following vehicle will collide with the followed vehicle. Thus, it is essential to consider this case when calculating the ACDA.

Assuming that vehicle B stops suddenly with zero stopping distance. Thus, setting $V_{B}=0$ in the previous equations yields:

In the case when the following vehicle does not have an AEB system:(8)$$V_{Amax}=\sqrt{{\left(fgT_{rA}\right)}^{2}+2gfD}-fgT_{rA}$$
(9)$$D_{min}=V_{A}.\;T_{rA}+\frac{V_{A}^{2}}{2gf}$$

In the case when the following vehicle has an AEB system ($T_{rA}=0)$:(10)$$V_{Amax}=\sqrt{2gfD}$$
(11)$$D_{min}=\frac{V_{A}^{2}}{2gf}$$

We present in Figure 20 the maximum speed of the following vehicle (A) gives the distance to avoid the collision, assuming that the followed vehicle (B) suddenly stops when the weather is clear and the case when the weather is foggy. In addition, the figure depicts the case when the vehicle is dotted with an AEB system for both dry asphalt and snowy road conditions. The maximum slowing speed for a given separation distance between the two vehicles is larger in the case when AEB is used compared to foggy and clear weather conditions. Here, we look at three (3) different scenarios on asphalt and snowy roads: the first scenario is when there is foggy weather (with driver reaction time) and in clear weather (with driver reaction), and the last scenario is when the vehicle has autonomous emergency braking; it can be observed that a vehicle with AEB system has the highest speed on both asphalt and snowy roads and when the vehicle is traveling in clear weather. However, the vehicle traveling in clear weather on asphalt has a higher speed than a vehicle with AEB system traveling on a snowy road. However, the maximum speeds are reduced significantly compared to the case when the followed vehicle normally stops (does not stop suddenly), as shown in Figure 14.

In Figure 21, we present the minimum allowed distance (in meters) for a given speed of following vehicle A (km/h) to avoid the collision if the followed vehicle B suddenly stops both on asphalt and snowy roads. The distance is shorter for a vehicle with AEB system traveling on asphalt (around 80 m) than a vehicle traveling in clear weather with driver reaction on asphalt, etc. The same scenario is seen on a snowy road. In this case, the minimum allowed safe driving distance is reduced compared to the case when the followed vehicle does not suddenly stop as shown in Figure 16 and Figure 17.

### 4.5. Mass of the Vehicle

In this section, we present the effect of the mass of the vehicle on the braking distance. From Newton’s Second Law of motion, the force (braking force since the vehicle concerned is already in motion) is the product of mass and acceleration (or, in this case, deceleration). The braking distance of a vehicle increases proportionally with its mass. Assume a force of 1500 N is applied by the brakes to stop a moving car with a mass of 750 Kg, the car will decelerate at 2 m/s^2^, but if the same force (F = 1500 N) is applied to stop a moving car with a mass of 1500 Kg, it will decelerate at only 1 m/s^2^. When the car’s mass is doubled and the same braking force is applied, the car is now slowing down at half rate. The car with double mass will take twice as long to come to a complete halt.

The braking distance (m) and mass of the vehicle (Kg) are described in Figure 22 at different speeds (km/h). At a given speed, the braking distance required for a vehicle to come to a complete halt varies directly with the mass of the vehicle. For example, a vehicle moving at 140 km/h can stop within a 50 m braking distance if it has a mass of less than 1000 Kg, whereas it requires around 375 m if it has a mass of 2000 Kg. A car moving with 20 km/h can halt at 50 m braking distance if its mass is 1500 Kg and needs only 25 m (half of the previous) to halt if it has 1250 Kg and so on.

The speed of the vehicle (m) and the braking distance (m) at different masses of the vehicle (Kg) are presented as shown in Figure 23. At a given mass, the braking distance directly correlates with the speed. A lighter vehicle requires a shorter braking distance compared to a heavier vehicle moving at the same speed. For example, if a 100 m braking distance is given, a 500 Kg car will speed up to around 140 km/h and still be able to brake safely, but a heavier car (say 10,000 Kg) will only speed at around 100 km/h for it to avoid the collision.

### 4.6. Required Braking Force Given the Mass and the Speed to Yield the Same Braking Distance

The effect of braking force in the determination of braking distance is discussed in this section. For a moving vehicle to come to rest, there must be work done sufficient to remove all its kinetic energy. Work done = Kinetic Energy, where work done is the work needed (in Joule) to remove all vehicle’s kinetic energy.

Kinetic Energy = 0·5 × mass × velocity^2^ measured in Joule, and the work done is the product of braking force and distance. Thus:(12)$$F\times D=\frac{m\times V^{2}}{2}$$
(13)$$D=\frac{m\times V^{2}}{2\times F}$$
(14)$$V=\sqrt{\left(\frac{2\times F\times D}{m}\right)}$$

Therefore, for a fixed maximum braking force, the braking distance is proportional to the square of the vehicle’s speed. In Figure 24, the required braking force (N) as a function of the vehicle’s mass (Kg) for different speeds (km/h) and different braking distances (m) are presented. For example, at 30 km/h, a 500 Kg vehicle needs around 1900 N to stop at a braking distance of 10 m, around 900 N to stop at 20 m, and 500 N to stop at 30 m. At a speed of 60 km/h, a 750 Kg vehicle needs 5000 N to stop at 20 m and 2500 N to stop at 40 m, and 1000 N to stop at 100 m, whereas at a speed of 90 km/h, a 2000 Kg vehicle needs a force of 20,000 N to stop at 30 m and around 8000 N for the same vehicle to stop at 100 m. We can deduce that at high speed, the required baking force is proportional to the mass of the vehicle and the square of the speed but inversely proportional to the braking distance.

## 5. Conclusions and Future Directions

In this paper, we thoroughly studied the minimum safe driving separation distance or the Assured Clear Distance Ahead (ACDA) that must be used to avoid forward collisions. We have presented the braking and the stopping distance as well as the braking dynamics that are essential to understand the minimum required separation gap to avoid the collision. We have discussed all factors that affect the ACDA, namely, the speed of both vehicles, the initial separation distance, the tire state, the weather condition, the road condition, and the vehicle’s mass. Furthermore, we have presented the necessary equations required to calculate that distance. In addition, we have devised and analyzed the maximum speed of the following vehicle to avoid the collision given the speed of the followed vehicle and the current separation distance for all studied factors that affect the stopping distance.

Moreover, both ACDA and the maximum safe driving speed equations and analysis were presented in the case when the followed vehicle suddenly stops. The study was complemented by the case when the following vehicle is equipped with Autonomous Emergency Braking (AEB) system. Finally, we have presented the required braking force as a function of the vehicle mass for different speeds and braking distances. We have presented the traditional and smart distance measurement techniques between the followed and the following vehicles that can be used in connected and autonomous vehicles. The presented analytical results are essential for connected vehicles and self-driving vehicles to avoid collisions and maximize road utilization, and hence reduce traffic problems. One important advantage of this study is using vehicle-to-vehicle communication of IoV; both vehicles know the factors affecting the ACDA and the maximum allowed safe speeds, and they exchange them; thus, each can calculate them in real-time.

The future directions of this study are to use the proposed calculations of ACDA and the safe following driving speed to detect automatically and autonomously tailgating violations using IoV. We are currently exploring this research direction.

## Figures and Tables

**Figure 1 sensors-22-07051-f001:**
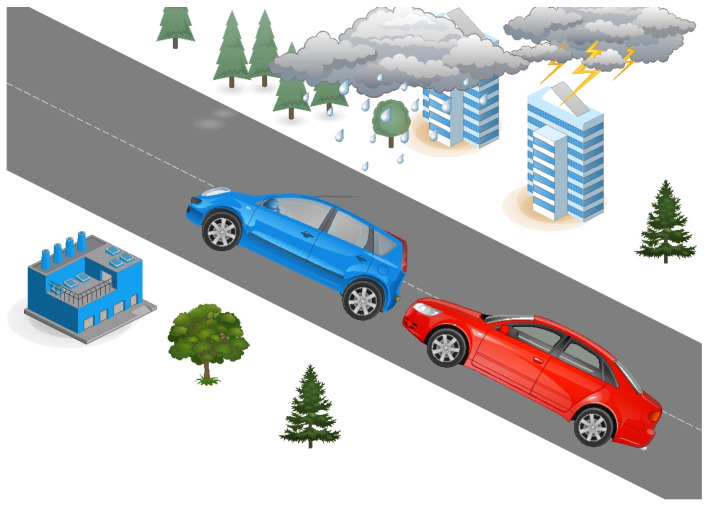
Vehicle tailgating. The red vehicle tailgates the blue one.

**Figure 2 sensors-22-07051-f002:**
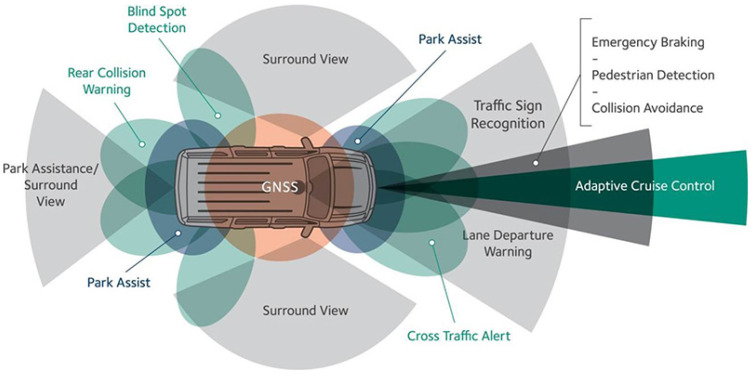
Different types of distance sensors in intelligent vehicles.

**Figure 3 sensors-22-07051-f003:**
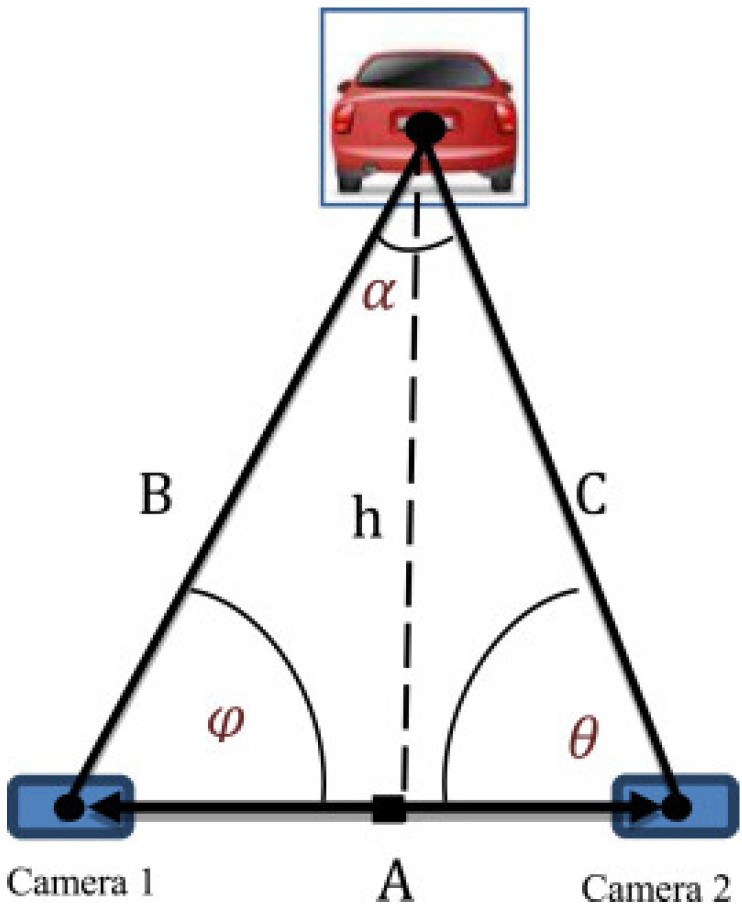
Stereovision system for distance estimation [52].

**Figure 4 sensors-22-07051-f004:**
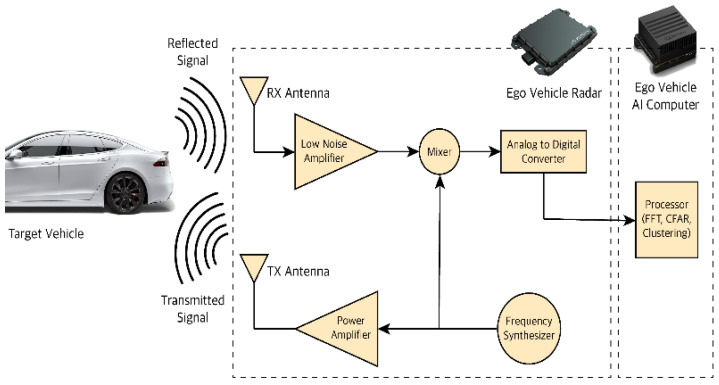
RADAR Distance measurement [54].

**Figure 5 sensors-22-07051-f005:**
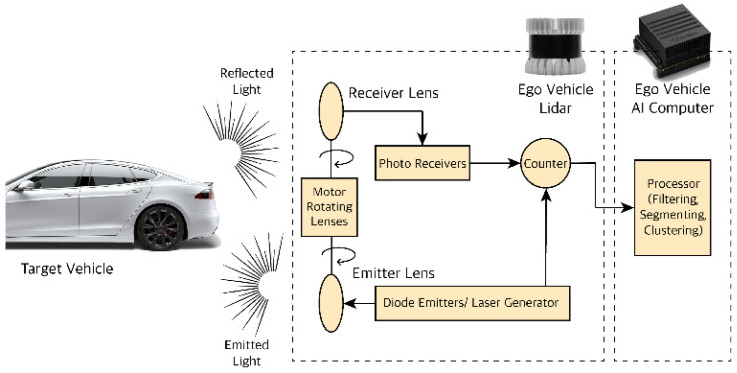
LiDAR Distance measurement [54].

**Figure 6 sensors-22-07051-f006:**
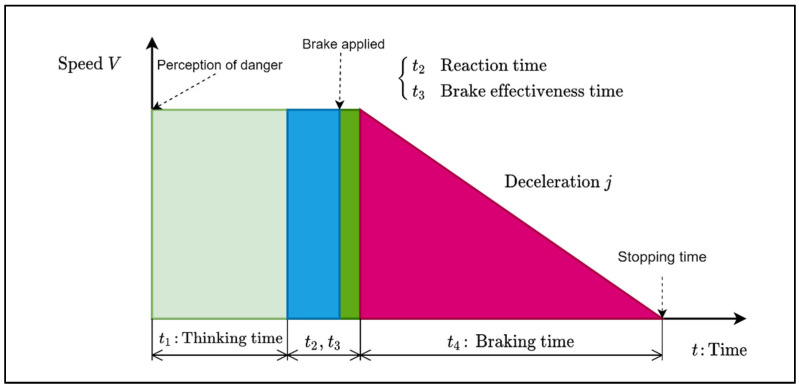
Stopping distance dynamics when the driver perceives a danger.

**Figure 7 sensors-22-07051-f007:**
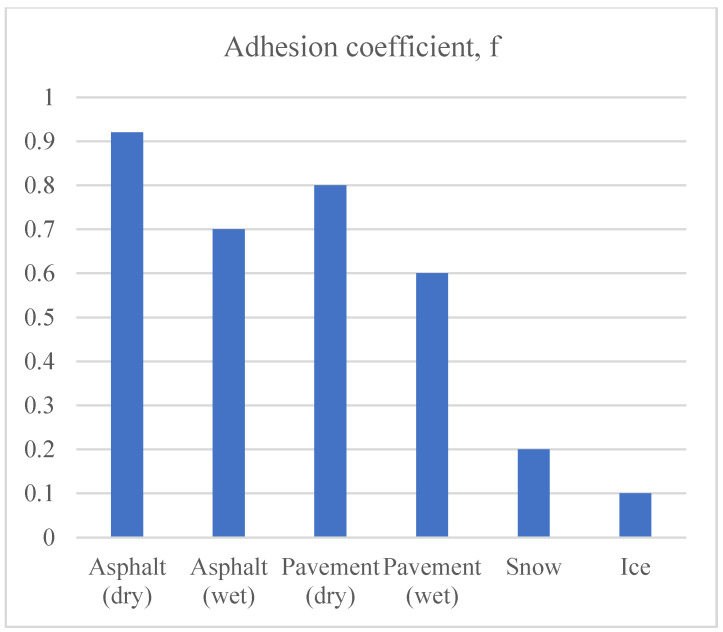
Adhesion coefficient (*f*) for different road types/conditions.

**Figure 8 sensors-22-07051-f008:**
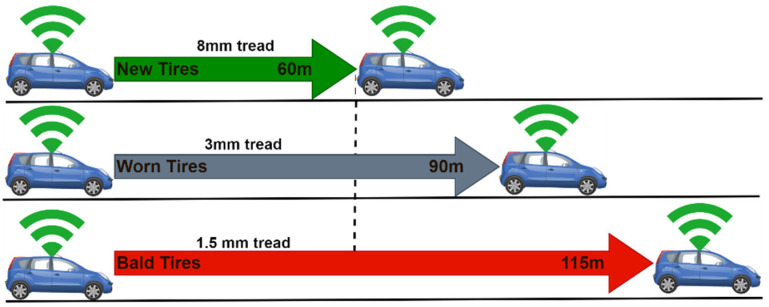
Effect of the tire state on the braking distance.

**Figure 9 sensors-22-07051-f009:**
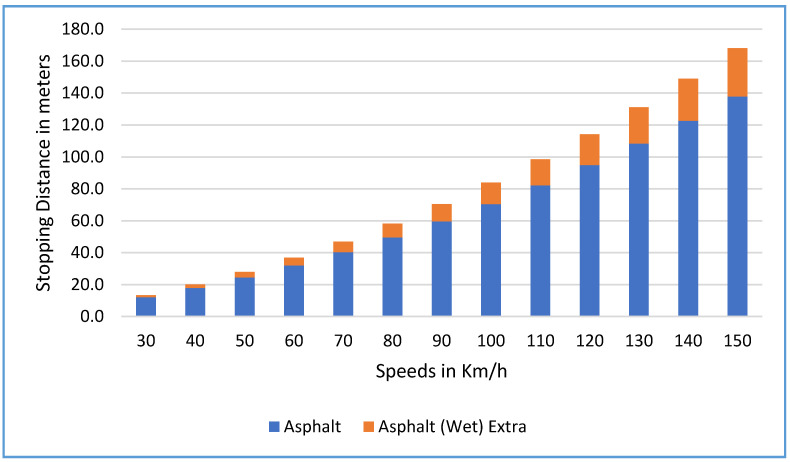
Extra stopping distance in meters on wet asphalt road compared to dry asphalt road for different speeds.

**Figure 10 sensors-22-07051-f010:**
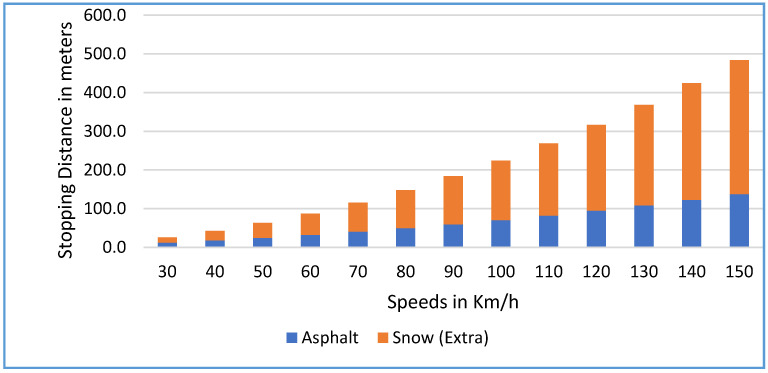
Extra stopping distance in meters on snow road compared to dry asphalt road for different speeds.

**Figure 11 sensors-22-07051-f011:**
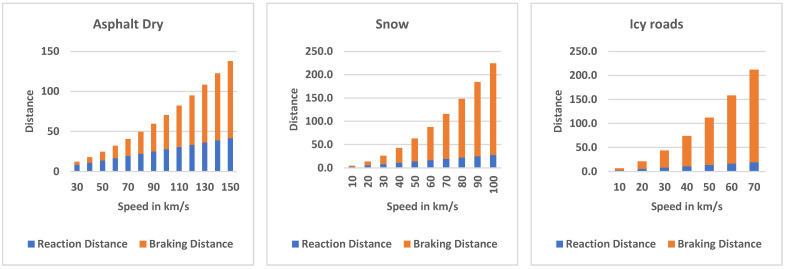
Stopping distance components as the sum of the reaction and braking distances for different speeds and on different road types. The reaction time is considered one second and assumes new normal tires.

**Figure 12 sensors-22-07051-f012:**
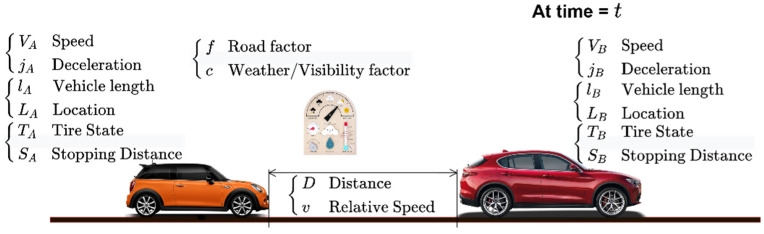
Parameters affecting the sufficient safe driving gap provided that IoV is used to broadcast these parameters to the surrounding vehicles in real time.

**Figure 13 sensors-22-07051-f013:**
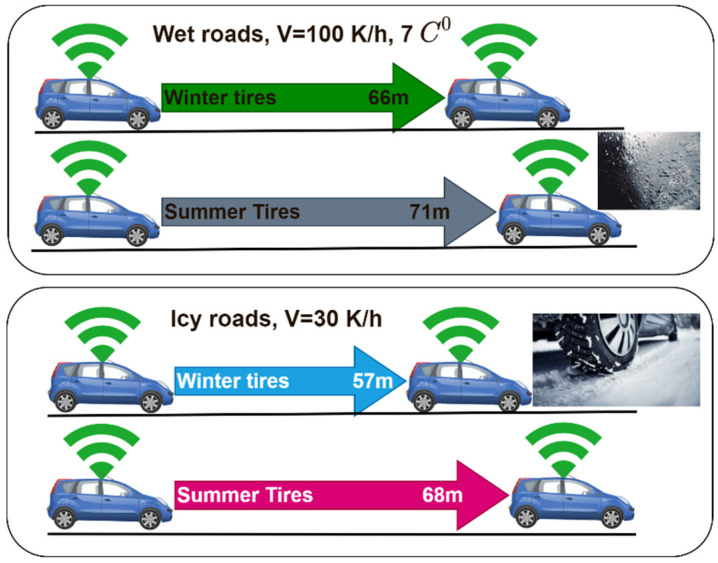
Effects of weather conditions and tire type.

**Figure 14 sensors-22-07051-f014:**
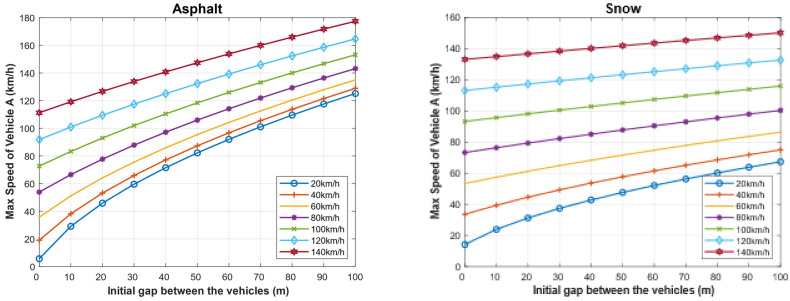
Maximum speed of the following vehicle for different speeds of the followed vehicle for different initial distances between the two vehicles when the driver reaction ($T_{rA}=1\;s$).

**Figure 15 sensors-22-07051-f015:**
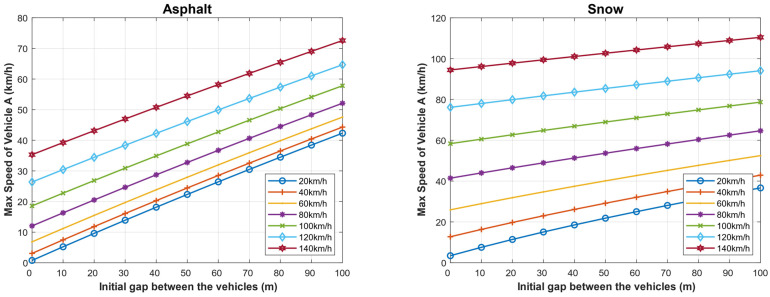
In foggy weather: maximum speed of the following vehicle for different speeds of the followed vehicle for different initial distances between the two vehicles when the driver reaction ($T_{rA}=8\;s$).

**Figure 16 sensors-22-07051-f016:**
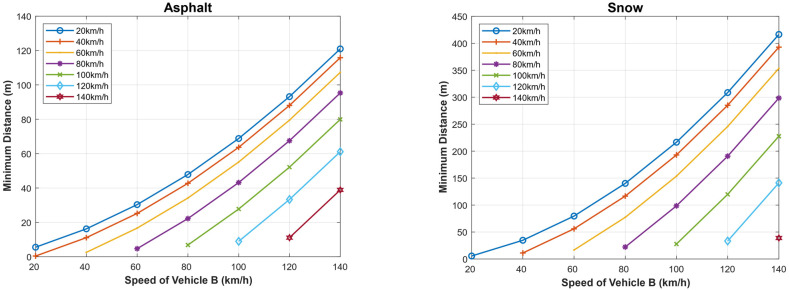
Minimum safe driving distance for different speeds of both the following and the followed vehicle in the case when the weather is clear, assuming the driver reaction time is 1 s for dry asphalt and snowy roads.

**Figure 17 sensors-22-07051-f017:**
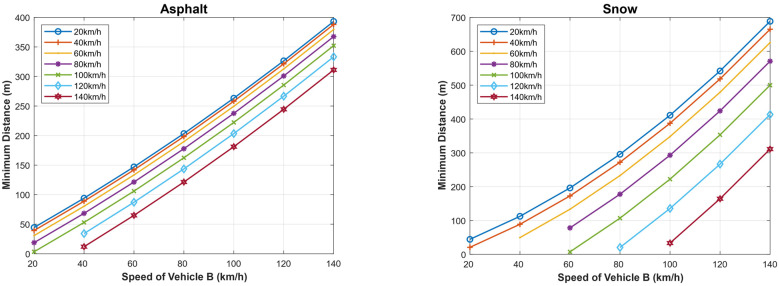
Foggy weather: minimum safe driving distance for different speeds of both the following and the followed vehicle in the case when the weather is clear, assuming the driver reaction time is 8 s for dry asphalt and snowy roads.

**Figure 18 sensors-22-07051-f018:**
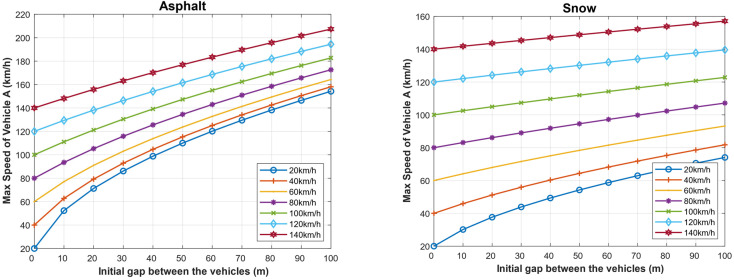
Maximum speed of the following vehicle for different speeds of the followed vehicle for different initial distances between the two vehicles when the following vehicle is dotted with Autonomous Emergency Braking (AEB) system ($T_{rA}=0\;s$).

**Figure 19 sensors-22-07051-f019:**
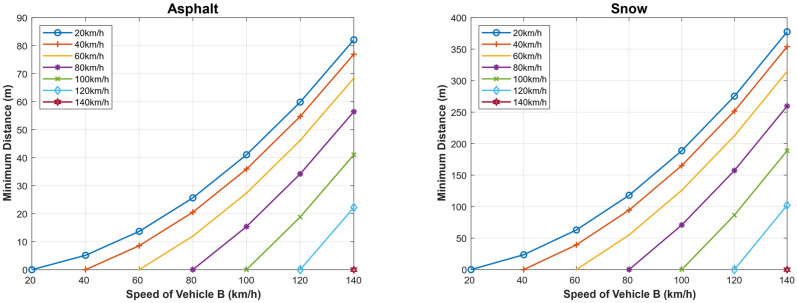
Minimum safe driving distance for different speeds of both the following and the followed vehicle in the case when the following vehicle is dotted with Autonomous Emergency Braking (AEB) sys-tem ($T_{rA}=0\;s$) for dry asphalt and snowy roads.

**Figure 20 sensors-22-07051-f020:**
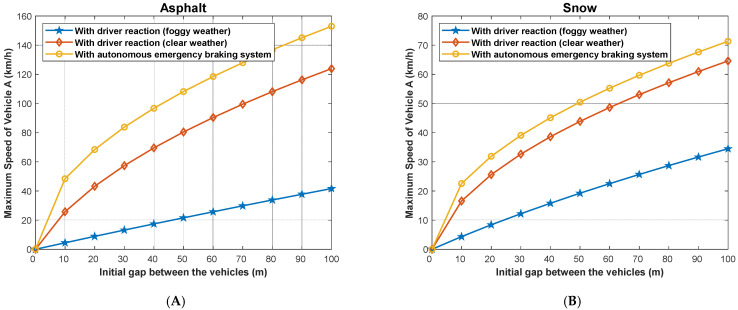
Maximum speed of the following vehicle (**A**) given the distance to avoid a collision, assuming that the followed vehicle (**B**) stops suddenly.

**Figure 21 sensors-22-07051-f021:**
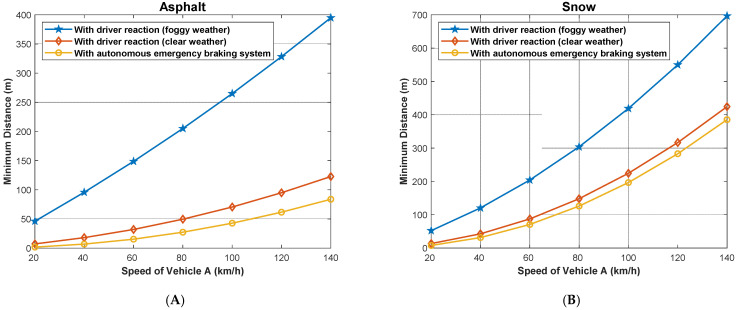
Minimum allowed distance for a given speed of the following vehicle (**A**) to avoid a collision, assuming that the followed vehicle (**B**) stops suddenly.

**Figure 22 sensors-22-07051-f022:**
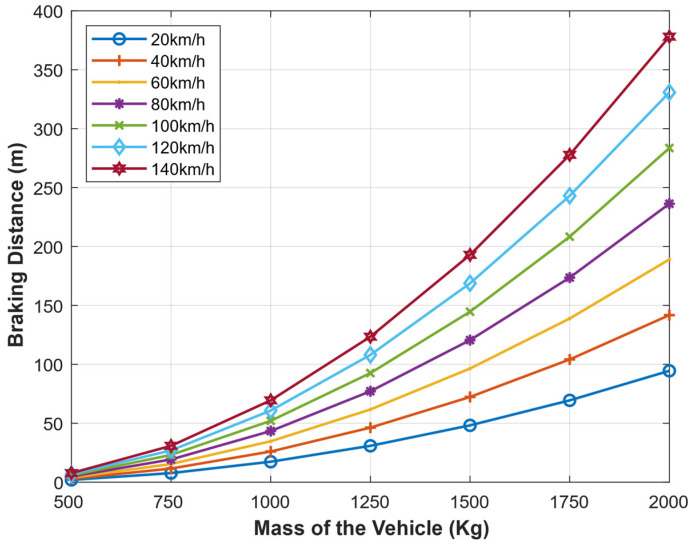
Braking Distance given the mass of a vehicle at different speeds.

**Figure 23 sensors-22-07051-f023:**
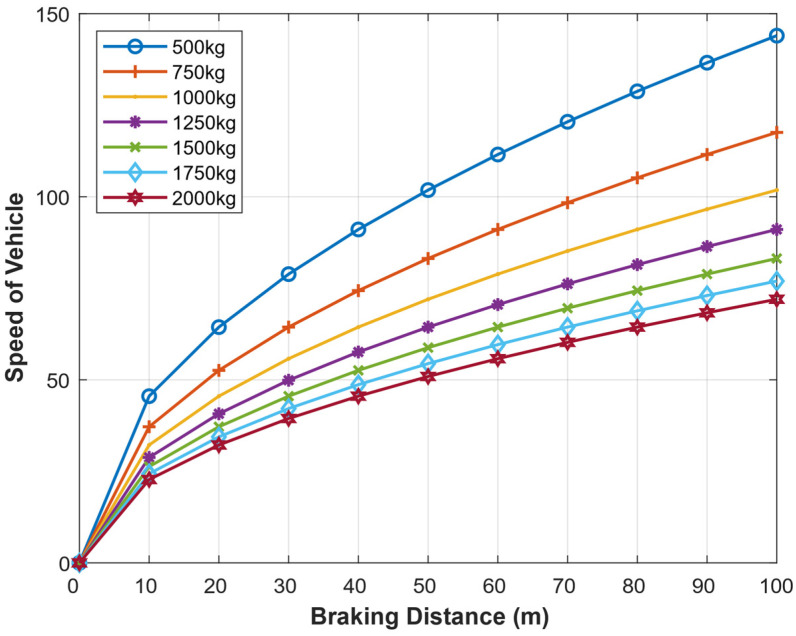
Maximum speed of a Vehicle given the braking distance with different masses of the vehicle.

**Figure 24 sensors-22-07051-f024:**
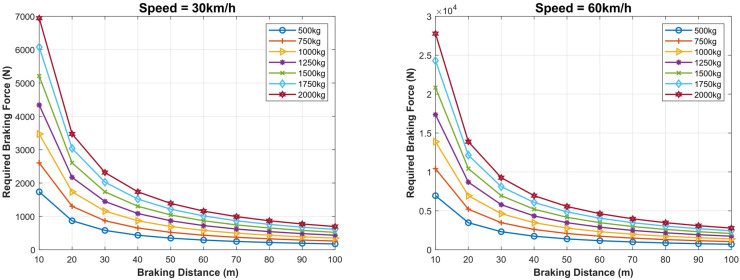
Required braking force as a function of the vehicle mass for different speeds and braking distances.

**Table 1 sensors-22-07051-t001:** Summary of the important related works and their limitations.

References	Considered Parameters	Category	Finding and Limitations of the Study
[32]	Speed and the deceleration of both vehicles.	Safe driving capacity	Only safe driving distance at the intersection and straight roads were considered as a function of the speed and deceleration. The study did not consider many important parameters such as the road stat, the current separation gap, the tires condition, the visibility, the weather conditions, the weight of the vehicles, the length of the vehicles, and the braking force. The study did not consider the case when the front vehicle stops instantly (in zero time). In addition, the study did not consider the effect of different driver reaction times. Additionally, the study did not consider the cases when the vehicle is equipped with an Autonomous Emergency Braking (AEB) system or not. The study did not consider the different types of distance measurement techniques used in IoV and CV.
[33,34,35]	Distance estimation using a camera.	Distance measurement	It does not consider any parameter of those considered in our proposed work. The work proposes an algorithm that uses a single camera to estimate the distance; it does not consider the safe driving distance or the safe driving speed.
[36,37]	Fog condition only	Safe driving distance	The study considers the car-following distance as a function of different levels of fog conditions; it does not consider any of the other important conditions that we are considering and mentioned in first row of this table.
[38,39]	Speed of the following vehicle and the distance	Safe driving distance	The study uses simulation and reinforcement learning to determine the safe driving distance as a function of the speed of the following vehicle and the separation gap only.
[34,40]	Distance estimation using two stereoscopic cameras.	Distance measurement	A stereovision-based approach for determining the safe driving distance. The proposed approach consists of having two cameras mounted on the security vehicle. The distance between the security vehicle and the ahead vehicle can be calculated using traditional camera calibration, and parameter distortion calculation. Although this approach is effective, it requires the presence of a security vehicle, which can be noticed by the driver. Furthermore, it is not suitable for next-generation ITS and connected vehicle technologies. In addition, it is just a measuring approach without considering the safe driving distance or speed.
[26]	Speed and gap only	Safe driving distance	A safety indicator called time gap interval for safe following distance is proposed, which incorporates vehicle dynamics and driver behavior factors, including the time component, to broadcast and propagate appropriate safety messages in a vehicular ad hoc network (VANET) environment. The study considered the car speed, the gap and the length of the vehicles only.

**Table 2 sensors-22-07051-t002:** Possible distance-measuring technologies that are used in vehicular systems. Data in this table is compiled from [49,50,51].

Technology	Minimum Range (m)	Maximum Range (m)	Resolution (mm)	Accuracy	Update Rate (Hz)	Minimum Field of View (deg.)
Mico/Short LiDAR	0.1	40	≈5	±5 cm	1–1000	≈4
Long Distance LiDAR	40	160	10	±10 cm	1–1000	≈0.5
Infrared Proximity Sensor	0.1	1.50	-	±1 cm	26	
Ultrasonic Range Finder	0.15	6.5	1	±1 cm	8–20	20–60
Stereo Camera	0.3	200	10	±5 cm	1–60	5–100
Standard GPS	3	∞	20	±300 cm	1–18	
Global navigation satellite system (GNSS)	1	∞	20	±100 cm	1–18	
Differential GPS (DGPS)	0.3	∞	20	±30 cm	1–18	
RTK	0.01	∞	1	±1 cm	1–20	

**Table 3 sensors-22-07051-t003:** Total stopping distance for different road types and for different speeds.

Speed	10	20	30	40	50	60	70	80	90	100	110	120	130	140	150
Asphalt	3.2	7.3	12.2	18	25	32	40	50	60	71	82	95	108	123	138
Asphalt (Wet)	3.3	7.8	13.4	20	28	37	47	58	71	84	99	114	131	149	168
Pavement	3.3	7.5	12.8	19	26	34	44	54	65	77	90	104	119	135	152
Pavement (Wet)	3.4	8.2	14.2	22	30	40	52	64	78	93	110	128	147	167	189
Snow	4.7	13.4	26.0	43	63	87	116	148	184	224	268	316	368	424	484
Ice	6.7	21.3	43.7	74	112	158	212	274	344	421	506	600	701	810	927

**Table 4 sensors-22-07051-t004:** Stooping Distance for different road types for different speeds. Reaction distance is not affected by the road type.

	Speed	20	30	40	50	60	70	80	90	100	110	120	130	140
(Km/h) Road Type														
Reaction Distance (Independent of the Road Type)		5.6	8	11	14	17	19	22	25	28	31	33	36	39
Asphalt	Braking Distance	1.7	4	7	11	15	21	27	35	43	52	62	72	84
Asphalt Wet	Braking Distance	2.2	5	9	14	20	28	36	46	56	68	81	95	110
Pavement	Braking Distance	2.0	4	8	12	18	24	31	40	49	59	71	83	96
Pavement Wet	Braking Distance	2.6	6	10	16	24	32	42	53	66	79	94	111	128
Snow	Braking Distance	7.9	18	31	49	71	96	126	159	197	238	283	332	385
Ice	Braking Distance	15.7	35	63	98	142	193	252	319	393	476	566	665	771

## Data Availability

The source code used to support the findings of this study is available from the corresponding author upon request.

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
