# Peer review of "Safe Driving Distance and Speed for Collision Avoidance in Connected Vehicles"

_sensors, 2022, doi:10.3390/s22187051_

Round 1
Reviewer 1 Report
1. The literature review is not written critically. It is not clear from the discussion what limitations exist in the present literature. Without this being done, the gap cannot be identified, and therefore, the motivation of the work is not clear.
2. The references cited in this paper are incomplete. The authors should also mention the relevant research in the recent literature. To name a few, the works of Wang, et al. (Wang, Y. Y., & Wei, H. Y. (2020). Road capacity and throughput for safe driving autonomous vehicles. IEEE Access, 8, 95779-95792.), Cao, et al. (Cao, Z., Yang, D., Jiang, K., Xu, S., Wang, S., Zhu, M., & Xiao, Z. (2019). A geometry-driven car-following distance estimation algorithm robust to road slopes. Transportation research part C: emerging technologies, 102, 274-288.), Wang, et al. (Wang, Z., Huang, H., Tang, J., Meng, X., & Hu, L. (2022). Velocity control in car-following behavior with autonomous vehicles using reinforcement learning. Accident Analysis & Prevention, 174, 106729.), and Huang, et al. (Huang, Y., Yan, X., Li, X., Duan, K., Rakotonirainy, A., & Gao, Z. (2022). Improving car-following model to capture unobserved driver heterogeneity and following distance features in fog condition. Transportmetrica A: Transport Science, 1-24.).
3. The parameter of the weather condition as shown in Figure 12 does not appear in the formulas presented in the manuscript. However, the authors listed the driver reaction time and the driving speed of the following vehicle under foggy weather in Section 4.2. It is not clear how driver reaction time and the driving speed is determined. Please clarify this.
4. Conclusions and Future Directions section should be corrected as Section 5.
5. In Conclusions and Future Directions section, the authors mentioned that “The presented results and experimentations are essential for the connected vehicles and self-driving vehicle development to avoid collision and to maximize road utilization, and hence reduce traffic problems”. However, it seems that only the analytical results are given in the manuscript. It is not clear how the experimentations were conducted in this work. The authors might need to clarity this.
Author Response
Response to Reviewer Comments #1
Dear respective reviewer,
We would like to thank you very much for your time, efforts, and constructive feedback and comments on our manuscript entitled “Safe Driving Distance and Speed for Collision Avoidance in Connected Vehicles”.
Here is a point-by-point response to your comments and concerns.
Comment#1: The literature review is not written critically. It is not clear from the discussion what limitations exist in the present literature. Without this being done, the gap cannot be identified, and therefore, the motivation of the work is not clear.
Response: Thank you for pointing this out. We agree with this comment. Therefore, we have reworked the related work section (Section 2). We have added a table-based summary of the most related studies and the limitations in these studies that have been tackled our study. In addition, many recent related works have been added to the references and reviewed in the related section part. In addition, we have added half a page to the Introduction section summarizing the contribution of our work in 6 items. Please refer to Section 1, Table 1 and Section 2 in the article for more details.
Comment 2. The references cited in this paper are incomplete. The authors should also mention the relevant research in the recent literature. To name a few, the works of Wang, et al. (Wang, Y. Y., & Wei, H. Y. (2020). Road capacity and throughput for safe driving autonomous vehicles. IEEE Access, 8, 95779-95792.), Cao, et al. (Cao, Z., Yang, D., Jiang, K., Xu, S., Wang, S., Zhu, M., & Xiao, Z. (2019). A geometry-driven car-following distance estimation algorithm robust to road slopes. Transportation research part C: emerging technologies, 102, 274-288.), Wang, et al. (Wang, Z., Huang, H., Tang, J., Meng, X., & Hu, L. (2022). Velocity control in car-following behavior with autonomous vehicles using reinforcement learning. Accident Analysis & Prevention, 174, 106729.), and Huang, et al. (Huang, Y., Yan, X., Li, X., Duan, K., Rakotonirainy, A., & Gao, Z. (2022). Improving car-following model to capture unobserved driver heterogeneity and following distance features in fog condition. Transportmetrica A: Transport Science, 1-24.).
Response: Thanks for your insightful comment. We have added these references and we added several other related references. Please refer to the references section and the related work section.
Comment 3. The parameter of the weather condition as shown in Figure 12 does not appear in the formulas presented in the manuscript. However, the authors listed the driver reaction time and the driving speed of the following vehicle under foggy weather in Section 4.2. It is not clear how driver reaction time and the driving speed is determined. Please clarify this.
Response: Actually, the weather condition (rainy, foggy, snowy or clear) are directly or indirectly included in almost the formulas. In rainy weather, the road condition will be wet and thus, the adhesive coefficient f=0.7 instead of .92 in the dry asphalt, as shown in Figure 7. Similarly, if the weather is snowy, then f=0.2. As shown in almost all the formulas, f is included (please refer to eq 2, 3, 4, 5, 6, 7, 8, 9 , 10). The only weather condition that affect the visibility is foggy weather. It is indirectly included in the formulas in the driver reaction time in the case when AEB is not used. We have added two recent related works justifying the correlation between the foggy weather and the driver reaction time. Several figures in the manuscript show the difference between the case of foggy weather and in clears weather for different road conditions.
Comment 4. Conclusions and Future Directions section should be corrected as Section 5.
Response: Thanks for the thorough revision. We have corrected the typo.
Comment 5. In Conclusions and Future Directions section, the authors mentioned that “The presented results and experimentations are essential for the connected vehicles and self-driving vehicle development to avoid collision and to maximize road utilization, and hence reduce traffic problems”. However, it seems that only the analytical results are given in the manuscript. It is not clear how the experimentations were conducted in this work. The authors might need to clarify this.
Response: We totally agree with your insightful observation. We have updated Section 5 to clarify this.

Reviewer 2 Report
Well, the manuscript-at-hand has spoken a lot about the notion of the 'Internet of Vehicles (IoV)' here and there, nevertheless, the same has not been formally introduced. I would suggest the authors to formally delineate on the notion of IoV and its essence for the futuristic Smart Cities. The authors may refer to the exisiting literature, i.e., on https://ieeexplore.ieee.org/document/9088328 and https://ieeexplore.ieee.org/abstract/document/9707170, for understanding the same.
The Contributions of the manuscript-at-hand should be documented in a much more categorical manner, i.e., towards the end of the Introduction as (1) ... , (2) ... , and (3) ... , and so on.
A Comparative Table in Seciton 2, Related Works, delineating the pros and cons of the referred approaches would be appreciated.
What is RTK? Please introduce the same once before using the Abbreviation.
Language of the manuscript-at-hand should be revisited carefully. Apart from the Sentence Structure, the Uniformity is important too, i.e., at some places, its wrtten Kg and other places delineate kg.
Simialrly, the size of the Figures (and the Subfigures) should be made uniform for a better Presentation. The Subfigure 2 of the Figure 19 (i.e., Snow) is smaller than the Subfigure 1 (i.e., Aspahlt). Also, Figure 22 and 23 are of different sizes.
Author Response
Dear respective reviewer,
We would like to thank you very much for your time, efforts, and constructive feedback and comments on our manuscript entitled “Safe Driving Distance and Speed for Collision Avoidance in Connected Vehicles”.
Here is a point-by-point response to your valuable comments and concerns.
Comment 1. Well, the manuscript-at-hand has spoken a lot about the notion of the 'Internet of Vehicles (IoV)' here and there, nevertheless, the same has not been formally introduced. I would suggest the authors to formally delineate on the notion of IoV and its essence for the futuristic Smart Cities. The authors may refer to the exisiting literature, i.e., on https://ieeexplore.ieee.org/document/9088328 and https://ieeexplore.ieee.org/abstract/document/9707170, for understanding the same.
Response: We totally agree with your insightful comment. We have updated the Introduction section by explaining more details about IoV. In addition, we have added the two references. Furthermore, we have reworked the related work section (Section 2). In addition, many recent related works have been added to the references and reviewed in the related section part.
Comment 2. The Contributions of the manuscript-at-hand should be documented in a much more categorical manner, i.e., towards the end of the Introduction as (1) ... , (2) ... , and (3) ... , and so on.
Response: Thank you for pointing this out. We agree with this comment. Therefore, we have added half a page to the Introduction section summarizing the contribution of our work in 6 items. Please refer to Section 1, page 3.
Comment 3. A Comparative Table in Seciton 2, Related Works, delineating the pros and cons of the referred approaches would be appreciated.
Response: Thanks for this fruitful comment. We agree that this will enhance the quality of the article. Therefore we have added the table in Section 2 (Page 4). We have updated (by adding more recent related works) Section 2.
Comment 3. What is RTK? Please introduce the same once before using the Abbreviation.
Response: Updated (page #6)! thanks for the feedback.
Comment 4 Language of the manuscript-at-hand should be revisited carefully. Apart from the Sentence Structure, the Uniformity is important too, i.e., at some places, its wrtten Kg and other places delineate kg.
Response: Thanks very much for the useful comment. We have proofread the manuscript and reviewed it intensively for typos, sentence structure, grammar, clarity and uniformity. We have checked all the units and fixed the issues.
Comment 5 Simialrly, the size of the Figures (and the Subfigures) should be made uniform for a better Presentation. The Subfigure 2 of the Figure 19 (i.e., Snow) is smaller than the Subfigure 1 (i.e., Aspahlt). Also, Figure 22 and 23 are of different sizes.
Response: The size and layout of all the figures were reviewed and adjusted. SubFigures have the same size now. Related figures have the size as well.

Round 2
Reviewer 2 Report
Thank you for incorporating the Recommended Changes.
The Overall Quality of the manuscript-at-hand has considerably improved.